# Frameworks of Movement Sciences

**Mitsumasa Miyashita**

Laboratory of Exercise Physiology and Biomechanics, Graduate School of Education, University of Tokyo, Honngo 7-3-1, Bunnkyo-ku, Tokyo 113-0033, Japan; mm1192@tcn-catv.ne.jp

**Abstract:** This article is composed of two parts. In the first part, a review is conducted on how research concerning human movement has been performed on Japanese subjects with newly developed methods in the last 60 years. In the second part, the frameworks of human movement sciences, such as exercise physiology, biomechanics, sports performance, and health, are proposed mainly based on the research results obtained by the author and his colleagues. It is expected that this article will be helpful to researchers in the fields of physical education, sports, and health.

**Keywords:** physical performance; physical resources; skill; biomechanics; exercise physiology

## 1. Introduction

The author is one of the pioneers of the field of human exercise sciences related to physical education, sports, and health in Japan [1]. The main topic of human exercise sciences is human movement.

Human movement is performed using chemical energy such as glycogen and oxygen through glycolysis and oxidation, respectively. Therefore, how much energy can be supplied by oxygen consumption and/or the amount of glycogen in muscles is an interesting topic of research. However, the methods to measure oxygen consumption and glycogen content in the past have been time-consuming and require specific skills. For instance, expired gas was collected by Douglas bags every minute, gas volume was determined by a dry gas meter, and samples of expired gas were analyzed for oxygen and carbon dioxide with a Scholander microanalyzer. Nowadays, oxygen consumption is determined by breath almost automatically.

On the other hand, human movements come in many forms and styles. Researchers have tried to study the kinematics and kinetics of human movements. For this purpose, the form or style was filmed on a 16 mm high-speed motion camera (Milliken Model DM55), which filmed at a normal rate of 500 frames/second. The resulting films were analyzed with the aid of an NAC Motion Analyzer, which enlarged the image 15 times and projected it onto an X–Y coordinates screen. To obtain the quantitative data from the film, the related points had to be detected frame by frame, so the process took a lot of time. At present, three-dimensional marker trajectories were recorded using a capture system at 200 Hz (Vicon, Oxford, UK). Using this system, the kinematics and kinetics of human movements were more easily studied compared to the past filming method.

The first half of this article consists of a reflection on the research activities conducted by the author and his research group in the past 60 years. The author and his colleagues had previously conducted their research using these older measuring methods. Then, racial differences and other findings in human movements were discussed, and several tools related to measuring methods were newly developed.

In the second half of this article, the frameworks of the human movement sciences are derived from the categories of the research activities and results obtained by the author and his colleagues.

More recently, advanced technologies have brought new aspects to the human exercise sciences. One example is the functional magnetic resonance imaging method, which reveals

the regional changes in cortical activation patterns during the performance of exercises [2,3]. Another example is the development of genetics, which reveals individual differences in responses to physical training [4].

The measuring methods of the human exercise sciences have been developed for the last several decades. The researchers have tried to investigate human movements by the contemporary measuring methods each day of their careers. This has followed the more accurate and precise knowledge that has been accumulated year by year.

This article aims to provide young researchers in the field of human movement with valuable insights and methodologies to enhance their scientific work in the future.

## 2. The Beginning of Research

### 2.1. Biomechanics of Swimming Fast

Since the author had been a swimmer on his university team, his first research interest involved the key factors in swimming fast. His trial aimed to conduct a biomechanical analysis of swimming, and the following three different aspects were clarified.

The swimmer who wants to swim fast is required (1) to take an ideal body position to reduce water resistance [5], (2) to minimize the fluctuation of swimming speed during a single-cycle stroke [6], and (3) to acquire a swimming style that produces fewer waves [7].

### 2.2. Why Recent Swimmers Swim Faster Than Past Swimmers

Referring to the results mentioned above and the changes in swimmers' environments, the reasons why recent swimmers can swim remarkably faster than the swimmers of 30–40 years ago are speculated as follows. Present swimmers have a large muscle mass developed through resistance training. They have higher swimming skills with small speed fluctuations because they started to practice swimming at a younger age than past swimmers. Moreover, the overflow of the current swimming pool sides is so flat that water easily flows over the wall to absorb the waves, and the floats between lanes are well-designed to absorb the waves; consequently, the present swimmers can swim faster. Also, the materials and designs of swimming suits have been improved to be less water-resistant.

## 3. Aerobic Power

### 3.1. Maximal Oxygen Uptake of Japanese People

Maximal oxygen uptake ($Vo_{2max}$) has been the main theme of exercise physiology. But, there were no data on Japanese people in 1965, when the author started to work as a researcher in exercise physiology. Therefore, the author and his colleagues challenged themselves to catch up with and become ahead of the researchers in Western countries.

The results revealed no racial difference in $Vo_{2max}$. The annual reduction rate of $Vo_{2max}$ in Japanese people was approximately similar to that of previous reports in Western countries [8,9]. A gender difference in the $Vo_{2max}$ of Japanese was found at all ages. The distribution of $Vo_{2max}$ in a given group of gender and age showed a normal distribution curve [9]. The $Vo_{2max}$ of both boys and girls increased with age from 12 years to 18 years [10].

### 3.2. Trainability of Aerobic Power

Physical exercise has been advocated for preventive, diagnostic, and/or rehabilitative purposes to minimize the many diseases associated with a lack of physical exercise, such as coronary heart disease, etc., in developed countries. Almost 10 years later than in Western countries, hypo-kinetic diseases had become one of the main research subjects in Japan.

Seven middle-aged females (23~40 years old) participated in a 44-week training experiment. Training intensities increased progressively from 60, 75, and 90% $Vo_{2max}$, keeping the total amount of work to 9000~12,000 kpm per day. Though individual differences were observed in the effect of training intensity on $Vo_{2max}$, the mean value of $Vo_{2max}$ definitely increased from 29.36 to 39.39 mL/kg/min [11].

### 3.3. Detraining Effect on Aerobic Power

It is well known that when training stops, cardiovascular and muscular fitness returns to pretraining revels. Eleven sedentary middle-aged (35~54-year-old) males volunteered to participate in the training experiment. The subjects walked 10 min at a constant speed of 110 m/min and at a constant slope, which was determined by pretest to be equivalent to 80% $Vo_{2max}$. The training was performed three days per week for 15 weeks.

The first test before training, the second test just after training, and the third test six months after the end of training were performed, respectively. The results showed that the mean values of $Vo_{2max}$ per body weight increased +11.7% by training and decreased −7.2% at six months after the end of training.

The gain in $Vo_{2max}$ acquired during training had not disappeared completely even after six months of sedentary life. Those results indicate that the level of physical activity itself in sedentary life might have changed for six months after the end of training [12].

### 3.4. A New Measure of Aerobic Power and New Bicycle Ergometer

$V_{O2max}$ has been widely used as the precise index of physical work capacity (PWC). But the measurement is complex and demands expensive research equipment and trained technicians. In addition, maximum effort is required for subjects. A new method to evaluate individual PWC, judging from the workload (WL) at which heart rate (HR) response to exercise reached the level of 75% of maximal HR ($PWC_{75\%HRmax}$), was proposed. In order to ascertain the validity of this $PWC_{75\%Hrmax}$, the correlation between $V_{O2max}$ and $PWC_{75\%Hrmax}$ was determined for 19 males and 4 females. The relationship between $Vo_{2max}$ and $PWC_{75\%Hrmax}$ showed a highly significant linear correlation [13].

Thereafter, a new bicycle ergometer measuring $PWC_{75\%Hrmax}$ was produced (Figure 1).

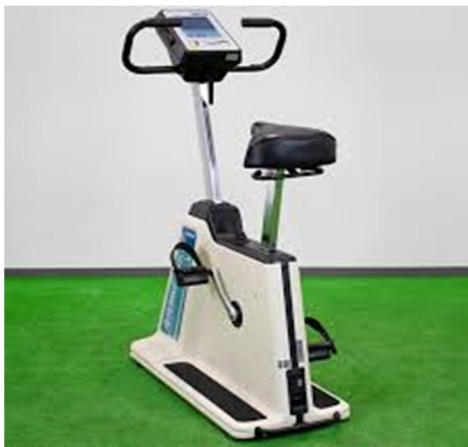

**Figure 1.** New bicycle ergometer for $PWC_{75\%HRmax}$.

A new bicycle ergometer utilized the electro-magnetic friction-braking system that could provide a high-speed control of WL in response to rapid HR changes during exercise. Therefore, WL in the new bicycle was automatically increased every 3 min to the suitable load according to the subject's HR responses. The HR biofeedback system in which R-R intervals are detected by a photoelectric pulse wave meter are equipped with the ergometer computes HR and the digital value of HR. Thus, with this ergometer, it is possible to safely test aerobic capacity from high-fit to low-fit laypersons with neither difficulty nor any assistance.

Extensive measurement of $PWC_{75\%Hrmax}$ was performed using this new bicycle ergometer on 788 healthy Japanese males and females between the ages of 20 and 68 years. Based on these findings, the evaluation criteria, including six-grade classifications for $PWC_{75\%Hrmax}$, were created [13].

## 4. Anaerobic Power

Several research results concerning the aerobic power of the Japanese have been reported in international journals and international symposiums. As a next step, the anaerobic power was investigated. Anaerobic power includes muscular static strength and dynamic peak torque exerted for a short period [14].

An interesting finding is that there are velocity-specific training effects derived from slow or fast contraction, and an intermediate training velocity may exist, which can enhance muscular power output over a wide range of contraction velocities [15].

### 4.1. Determination of Maximal Power Output

Maximal power output during short-term exercise has been determined under various conditions, such as climbing stairs, vertical jumping, and pedaling on a cycle ergometer. Since maximal power output varies depending on the loads, such as body weight, effect of gravity or external loading, and resistance, a new method to estimate the maximal anaerobic power determined by performing maximal cycling exercises was developed (Figure 2).

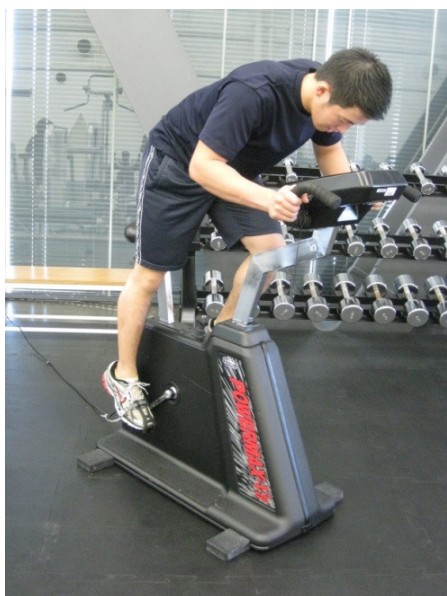

**Figure 2.** New bicycle ergometer for maximal anaerobic power.

Twenty-six male subjects pedaled the Monark bicycle ergometer as rapidly as possible at eight different loads from 18.7 to 84.2 Nm for 10 s, respectively. The wheel revolution rapidly reached the minimal value after starting to pedal, remained constant, and increased slowly again.

The minimal revolution time was adopted for the calculation of the maximal pedaling rate of eight loads. Since there was a negative linear relationship between pedaling rates in rad/s and loads in Nm, a linear regression equation was determined on all subjects; the correlation coefficient ranged from $-0.976$ to $-0.999$.

The power exerted during maximal pedaling can be determined from the pedaling rate at a corresponding load; pedaling rate x load. Then, three loads were selected from light load to heavy load for estimating the maximal anaerobic power by automatically detecting pedaling rates to respective loads [16].

### 4.2. Anaerobic Threshold of Japanese Athletes

The so-called anaerobic threshold (AT) is determined as a marked increase in blood lactate and/or the associated alternations in gas exchange parameters during an incremental exercise. Using the methods of incremental workload, we measured the values of AT in Japanese.

Ten male and eight female college distance runners performed an incremental running test on the treadmill. $Vo_{2max}$, lactate threshold (LT), and respiratory compensation threshold (RCT) were determined by the test. Though LT was not correlated to the running performance, RCT was highly correlated to the running performance [17].

The $Vo_{2max}$, aerobic threshold (AerT), and anaerobic threshold (AnT) were determined during a progressive bicycle ergometer on 24 elite speed skaters. A significant difference between the top 10 elite skaters and other skaters was found only in $Vo_{2max}$ but not in AerT and AnT [18].

## 5. Motor Skills

There are specific skills for various movements and appropriate ages at which the motor skills can be easily acquired. The skills conducted in various movements and the developmental aspects of movement patterns in young children were investigated.

### 5.1. Development of Manipulative Skill

The manipulation development of Japanese children aged two to six years was studied, using drawing with a pencil as a performance skill. Manipulation of pencil develops as follows; (1) from the most immature palmar grip stage when a drawing is made chiefly by the motion of the arm and shoulder; (2) progresses through the period when children try various finger postures; (3) they acquire the tripod; and (4) finally the small, highly coordinated movements of the fingers are possible at about four years of age. The grasping position moves progressively from the upper part to the lower part of the pencil.

A comparison of British and Japanese children shows that Japanese children are more advanced than British children at around three years of age in the manipulation of a pencil. It is suggested that these differences can be attributed to cultural factors [19].

### 5.2. Development of Throwing Skill

In comparison with inborn movements such as walking or running, many technical forms of throwing are apparent, ranging from nonskilled to skilled.

The developmental progress of throwing movements as related to age and sex was investigated, focusing mainly on the ball speed and throwing distance. Overhand ball-throwing movements on 180 boys and girls aged three to nine years were observed using a high-speed motion camera. No sexual differences were found in initial ball speed and throwing distance at three to four years of age. However, after around the age of five up to nine years, the speed of the ball and throwing distance rapidly increased in boys, while in girls, the ball speed did not. Thus, sex differences in throwing ability between the ages of five to seven years were observed.

The reason why improvement of throwing ability was seen only for boys was speculated as follows; in Japan, baseball is popular and televised every day, and most fathers tend to play "catch ball" with their sons. In addition, children of ages five to six years play with members of the same sex. Such sociological factors would facilitate an increase in skill level for the boys [20].

### 5.3. Skills of Other Movements

In general terms, common elements in motor skills are reproducible across similar movements. The trajectories of bowling balls thrown by three bowlers were analyzed on different levels of average scores. The proficient bowler's ball showed the same trajectories 10 times, while the poor bowler's ball showed large different trajectories among 10 trials [21].

Coordination of joints of the upper extremity was observed while throwing a ball and tennis serving. The overhand throwing motion of a trained baseball pitcher and the tennis serving motion of a highly trained tennis player were filmed using a high-speed motion camera. The apparent factor in distinguishing the level of movement skill was the relative position of the elbow joint in both movement patterns. The trained pitcher during

throwing and/or the trained tennis player during serving extended the elbow during the forward swing. (1) The elbow was pulled forward, (2) the forearm was whipped forward, and (3) the wrist was flexed explosively. The present study indicated that highly intensive training for the delivery of a faster ball might increase the high power of the muscular contraction per se and also improve the sequence and timing of muscle activities [22].

The grip firmness did not increase the ball speed after impact in tennis [23]. The mechanical responses of the tennis racquet frame, such as vibration after impact or bending forces acting on the frame during the swing, were investigated. The results indicated that the skilled tennis players almost hit the ball in the sweet spot of the racquet, which is swiftly moved [24].

*5.4. Effect of Running Skill on Long-Distance Running Performance*

It is generally accepted that $Vo_{2max}$ is a key factor in long-distance running. But other factors such as the fractional utilization of $Vo_{2max}$, muscle fiber composition, peak muscle and blood lactate accumulation, and running economy might affect long-distance running performance.

Then, how much the running economy affects the running performance was studied. Although there was a high linear relation between the mean speed of 5000 m run and $Vo_{2max}$ per body weight among 45 trained runners aged 19–23 years, the mean speed was largely different among the runners with a similar $Vo_{2max}$.

In comparing runners whose $Vo_{2max}$ was about 70 mL/kg/min, some runners ran at a mean speed of about 5.2 m/s, while other runners ran at a mean speed of about 5.6 m/s. Using a high-speed motion camera, the running styles in a single stride were analyzed while running 5000 m. The results showed that the slower runners utilized their energy to perform more vertical work than the faster runners [25]. In other words, excellent runners can run more economically (higher skill) than the non-excellent ones.

On the other hand, it was regretful how senescence deteriorated physical performance in swimming, running, and other activities could not be revealed.

## 6. Exercise Training for Competition

### *6.1. Strength Training*

The effects of strength training were evaluated on the performance of competitive swimmers. For instance, eight swimmers performed the training program of isokinetic, isotonic, and swim training six days per week for ten months. All subjects improved their mean speed of 50 m maximal swimming and exceeded their previous best times. A significant correlation was found between the gain in speed of 50 m swimming and muscular endurance [26].

Isokinetic peak torque and muscle fatigability in knee extensors of 203 junior speed skaters (aged 10 to 18 years) were tested in comparison with comparatively sedentary age-matched control subjects. The results indicated that after pubescence, speed skating training programs facilitated the morphological and functional development of the knee extensors [27].

Thus, the top level in sports requires athletes to be proficient in a number of physical performance attributes such as high levels of aerobic power, the ability to sprint, anaerobic power, strength, flexibility, and so on.

### *6.2. Fatigue Induced by Training*

The causes of fatigue during prolonged exercise, the responses of serum adrenocorticotropic hormone (ACTH), and cortisol or glucose ingestion were investigated.

Five subjects cycled at the same intensity (50% $Vo_{2max}$) until exhaustion or for up to 3 h before and after seven weeks of vigorous training. The results indicate that high-intensity training reduces the magnitude of the increase in serum ACTH and cortisol concentrations during prolonged low-intensity exercise, while the level of the blood glucose concentration did not change [28].

Seven young, healthy males were tested on the bicycle ergometer. Glucose or placebo was ingested before the test. It seems that a small amount (less than 50 g/h) of glucose ingestion neither changes carbohydrate metabolism nor improves the performance of intensive endurance exercise [29].

## 7. Exercise Training for Health

### 7.1. Exercise for Children

Physical inactivity has been a risk factor for many chronic diseases, such as type 2 diabetes, cardiovascular diseases, and certain cancers. Also, increases in obese and older persons have brought serious burdens on socio-economic aspects of our lives.

The effect of aerobic exercise of daily physical activity levels on the aerobic power in 12 preadolescent boys (9–10 years of age) was studied [30]. Daily physical activity levels were evaluated by a HR monitoring system for 12 h on three different days. The results showed that longer daily physical activities at moderate to higher intensity for preadolescent children seemed to increase aerobic power. Namely, habitual aerobic exercise in daily life plays an important role in maintaining aerobic power.

### 7.2. Sleeping and Heart Rate

The effect of eight weeks of submaximal aerobic exercise on asleep HR was examined in 12 sedentary middle-aged women and found that asleep HR in all subjects decreased significantly from 63.7 to 59.0 bpm, indicating that light exercise might enhance the quality of sleeping in middle-aged women, although the eye movements or EEG were not observed in parallel [31].

### 7.3. Aqua-Pole Walking

Most longitudinal studies have found that high levels of physical activity are associated with a reduction in the risk of cognitive decline and dementia. A new exercise was proposed, which might contribute to the prevention of cognitive decline and dementia as follows; the central motor cortex moves hands and arms in addition to feet and legs, while the primary sensory cortex receives stimuli from the hands and arms in addition to feet and legs. Thus, more cells of both motor and sensory cortexes are activated during walking with poles in the water.

As humans tend to lose their balance in the water, specially designed poles made of stainless steel were developed, which stand vertically in the water. In other words, the part of the pole with the grip always floats vertically in the water by buoyancy. The users can easily grasp the poles and keep their bodies in an upright and stable position. Aqua-pole walking is recommended for elderly persons in order to keep their walking abilities [32].

## 8. New Frameworks of Movement Sciences

As is introduced in the previous paragraphs, the author and colleagues have conducted various research on human movements making use of new methods and tools. Based on the research activities and results obtained, frameworks of human movement sciences are proposed.

### 8.1. Sports Biomechanics

Biomechanics has been used to describe the research on the mechanical aspects of human movement. Motion is the prime element in most human movements. The motion results in physical performance such as running time, distance of jumping, time of swimming, attractiveness of figure-skating, and so on.

Motion is created through force. There are two forces acting on the human body; a positive force generated by muscular contraction and passive forces produced by gravity, buoyance, resistance [5], reaction force [23,24], etc.

The muscular contractions are controlled by the nervous system according to the individual purpose of sports in Figure 3. For instance, baseball pitchers and tennis players

produce the fastball with the high power of muscular contraction and also improve the sequence and timing of muscle activities [22], while the passive force is only partially controlled by the nervous system. For instance, water resistance can be reduced by the swimmer by taking his body position in a stream line [7]. The muscular contraction is slightly modified through the proprioceptive reflex.

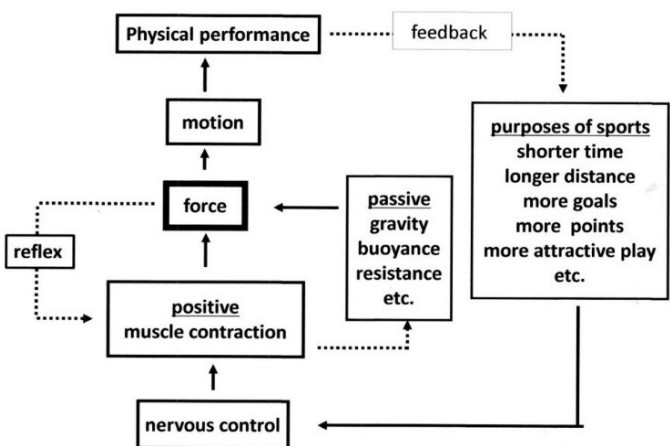

**Figure 3.** The framework of biomechanics.

Furthermore, through the feedback system, a more reasonable motion is acquired, referring again to the purpose of movement [21].

*8.2. Exercise Physiology in Addition to Biomechanics*

Through physical movement, a man converts chemical energy accumulated in the body into mechanical energy. The content of chemical energy in the body is expressed as physical resources (body size, $Vo_{2max}$, maximal oxygen debt, muscular strength, power, endurance, etc.).

On the other hand, mechanical energy results in physical performance. The matters concerning physical resources are studied mainly in exercise physiology, while the matters concerning physical performances are mainly in biomechanics.

Training increases $Vo_{2max}$, muscular strength, and others [11,14]. Exercise practice (exercise drill) improves physical performance. For instance, a bowling ball is thrown in the same way by a proficient bowler [21]. The results of both exercise physiology and biomechanics should be attributed to better methods of training and practice (Figure 4). Therefore, the researchers have to perform research on human movements from two aspects of exercise physiology and biomechanics and provide effective methods of exercise training and practice.

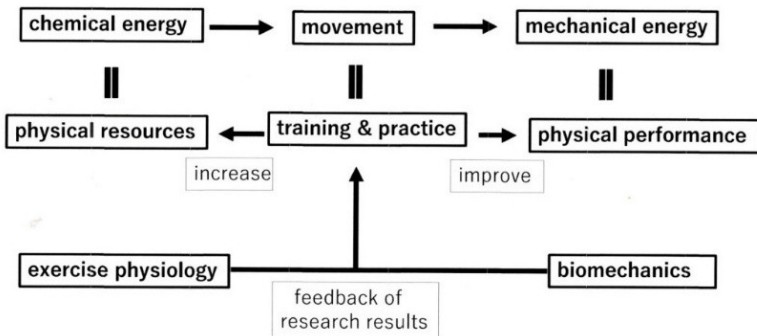

**Figure 4.** Human activity approached by exercise physiology and biomechanics.

### 8.3. Sports Performance

Sports performance reflects a player's abilities. One of them is fitness, in other words, physical resources. Physical resources are mainly composed of the cardiovascular and muscular systems.

The cardiovascular system develops with age up to 20–30 years [10] and declines with age after 30–40 years [8,9]. Physical growth increases physical resources, while senescence decreases physical resources. In contrast, detraining decreases the $Vo_{2max}$ [12]. A muscular system, such as static strength and dynamic power, develops with age [14].

Thus, sports performance depends fundamentally on the player's fitness, such as greater physical power and muscular potentials, but is considerably modified by the nervous function (so-called skill). Nervous functions develop at the young ages of two to six years [19] and seven to nine years [20].

Therefore, a sports player's ability is developed with age. Though there exist genetic limitations, mainly the player's ability is enhanced fundamentally by exercise training and practice [14,18]. Those exercises, training, and practice depend on facilities, advice from coaches, and scientific research results.

Finally, psychological and physical conditions may decide the win or loss of the competition on the day of competition (Figure 5).

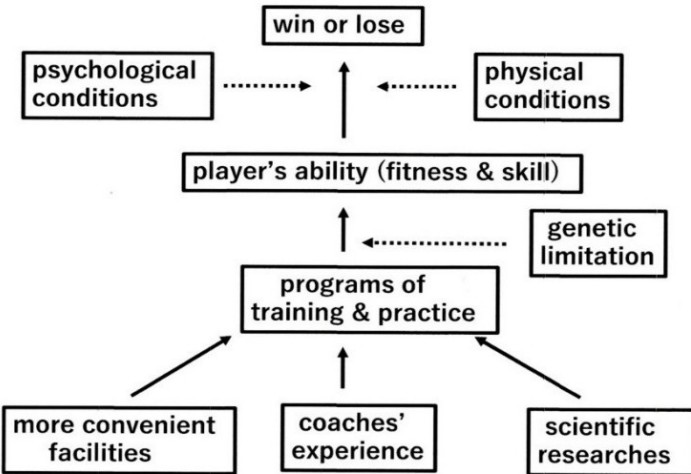

**Figure 5.** Competitive results in sports (win or lose) are affected by many factors.

### 8.4. Health Enhancement

The physiological studies investigating the effects of exercise training may be expected to play important roles in the health enhancement of laypersons. For instance, seven middle-aged females increased their $Vo_{2max}$ from 29.36 to 39.39 mL/kg/min on average after 44 weeks of training [11].

Thus, in the applied field of exercise physiology, researchers try to make use of the obtained results effectively for related people, including disabled persons [32].

Health tends to change from good to poor conditions regarding exercise, nutrition, sleep, aging, and other factors. For instance, in the case of preadolescent boys, longer daily activities tended to increase aerobic power [8]. In the case of middle-aged females, eight weeks of submaximal aerobic exercise decreased the nocturnal heart rate from 63.7 to 59.0 bpm [31].

Thus, daily proper exercise and adequate nutrition enhance health, while sedentary life and malnutrition deteriorate health. Moreover, senescence, accidents, and infection cause diseases. At that time, treatments might be needed; otherwise, death cannot be avoided (Figure 6).

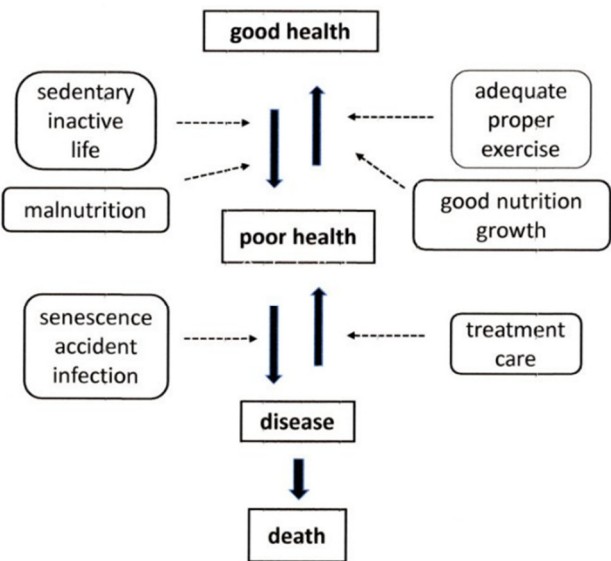

**Figure 6.** Health is changeable by daily physical activities.

### 9. Conclusions

For the past 10–20 years, the remarkable development of computer-controlled motion analysis has helped researchers of biomechanics to investigate human movements more precisely and rapidly than before.

Moreover, chemical biology, molecular biology, and genetics bring wider views to exercise physiology. Consequently, recent researchers have obtained much experimental data about the research target: human movements.

Therefore, the experiments conducted in both biomechanics and exercise physiology have supplied too much data for recent researchers to deal with. The coming researchers should carefully select and propose the research results to sports coaches, fitness instructors, and persons of rehabilitation who need research results to be applied to athletes and laypersons.

**Funding:** This research received no external funding.

**Informed Consent Statement:** Informed consent was obtained from all subjects involved in the studies cited in this article.

**Acknowledgments:** All co-authors of the articles cited in this manuscript agreed with this submission.

**Conflicts of Interest:** The author declares no conflict of interest.

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
