# Peer review of "Frameworks of Movement Sciences"

_applsci, doi:10.3390/app13148296_

Round 1
Reviewer 1 Report
The manuscript entitled "Frameworks of Movement Sciences" is very interesting and provides important information on the field.
However, there are several issues with the approach of the content. There are missing references in many parts of the article.
The manuscript is more a review, rather than an article type.
Author Response
Thank you very much for your valuable comments. I revised the manuscript and especially extended the content of introduction.Reviewer 2 Report
The article "Frameworks of Movement Sciences" by Miyashita Mitsumasa proposes frameworks for human movement sciences based on research activities performed on Japanese subjects over the last 60 years. The article is divided into two parts, with the first part reviewing how research activities concerning human movements have been performed on Japanese subjects with newly developed methods. This overview can be useful for researchers who are interested in understanding the evolution of research methods and techniques used in human movement sciences.
The second part proposes frameworks of human movement sciences such as exercise physiology, biomechanics, sports performance, and health based mainly on the research results obtained by the author and his colleagues.
The article begins by discussing how research activities concerning human movements have evolved over time and how new methods have been developed to study different aspects of human movement. The author then proposes frameworks for human movement sciences that can be used to study different aspects of human movement from multiple perspectives. These frameworks include exercise physiology, biomechanics, sports performance, and health.
The framework of exercise physiology focuses on understanding the physiological responses to exercise and how they affect physical performance. This includes studying factors such as body size, VO2max, maximal oxygen debt, muscular strength, power, endurance, etc. The framework of biomechanics focuses on understanding the mechanical aspects of human movement such as forces and torques that affect the body during movement. This includes analyzing movement patterns and joint mechanics to optimize performance or prevent injury.
The framework of sports performance focuses on optimizing athletic performance through training programs that target specific physical resources such as strength or endurance. This framework also includes studying factors such as nutrition and recovery strategies to enhance athletic performance and, at last, the framework of health focuses on understanding how physical activity affects overall health and well-being.
Furthermore, the article emphasizes the importance of interdisciplinary collaboration between different fields within human movement sciences. By combining knowledge and expertise from different fields, researchers can gain a more comprehensive understanding of human movement and its impact on physical performance and overall health.
I highlight a less positive aspect in this article, it does not provide specific information about newly developed methods used in research on human movements.
Overall, the article is a valuable contribution to science in general as it provides insights into how different frameworks can be used to study various aspects of human movement from multiple perspectives. It also highlights the importance of interdisciplinary collaboration and provides a historical overview of research activities concerning human movements.
Author Response
The reviewer indicated the manuscript did not show the newly methods. But many different methods cited will contribute to the young researchers.Reviewer 3 Report
This is submitted as an Article, and not as a literature review, as it seems it is. There are serious corrections to be done before it can be published. Please consider the suggestions below.
Abstract
1. The abstract does not clearly state the objective or purpose of the research. Although it mentions two parts of the article, it fails to clearly articulate the main goal or hypothesis of the research.
2. The abstract is not specific about what newly developed methods have been used over the last 60 years to study human movement in Japanese subjects.
3. Abstracts should ideally include a brief outline of the main results or findings of the study. This abstract mentions that it proposes frameworks based on research results but does not summarize what those results or proposed frameworks are.
4. it could also benefit from improved grammar and writing style for clarity. It reads somewhat awkwardly, which could impede understanding
Introduction
The introduction is too short. Please consider the following:
1. You mentioned research activities conducted over the past 60 years, but the introduction could benefit from more specific examples or a brief summary of some of the most significant works. This can give readers a more concrete understanding of the research topics that the manuscript will cover.
2. The introduction mentions racial difference and other findings in human movements. However, it's unclear what these findings are.
3. In the second part, it would be beneficial to mention the specific categories from which the human movement sciences frameworks are derived.
4. Instead of simply stating that young researchers may find useful information, it would be more engaging to directly address the reader. For example, you could write, "This article aims to provide young researchers in the field of human movements with valuable insights and methodologies to enhance their scientific works in the future."
Methods
This article does not follow the usual headers required for an original article, such as: Introduction/Background, Methods, Results, Discussion, Conclusions. Therefore, the methodology has serious flaws. Please revise
Results
This article does not follow the usual headers required for an original article, such as: Introduction/Background, Methods, Results, Discussion, Conclusions. Therefore, the methodology has serious flaws. Please revise
Discussions
This article does not follow the usual headers required for an original article, such as: Introduction/Background, Methods, Results, Discussion, Conclusions. Therefore, the methodology has serious flaws. Please revise.
Conclusion
Ok
References
Most of them are outdated. If this is a literature review, consider adding 30-40 more references. If you make this an Original article, update the references to have about 70% of them published after 2013.
There are moderate english spelling mistakes and errors.
Author Response
Thank you very much for your valuable comments. I revised the introduction referring to the three current articles, and added reviewer's comment [This article aims to provide young researchers in the field of human movements with valuable insight as methods develops to enhance their scientific work in future].Round 2
Reviewer 1 Report
Thank you for the revision and the effort on the manuscript.
Reviewer 3 Report
Thank you for accepting and doing the requested changes.